# Development of Negative Controls for Fc-C-Type Lectin Receptor Probes

Rémi Hatinguais,[a] Madalaine Kay,[b] Fabián Salazar,[a] Daniel P. Conn,[a] David L. Williams,[c] Peter C. Cook,[a] Janet A. Willment,[a,b] Gordon D. Brown[a,b]

[a]MRC Centre for Medical Mycology, University of Exeter, Exeter, United Kingdom

[b]Aberdeen Fungal Group, University of Aberdeen, Institute of Medical Sciences, Foresterhill, Aberdeen, United Kingdom

[c]Department of Surgery, James H. Quillen College of Medicine, Center for Inflammation, Infectious Disease and Immunity, East Tennessee State University, Johnson City, Tennessee, USA

Janet A. Willment and Gordon D. Brown equal contribution to senior authorship.

**ABSTRACT** Fc-C-type lectin receptor (Fc-CTLRs) probes are soluble chimeric proteins constituted of the extracellular domain of a CTLR fused with the constant fraction (Fc) of the human IgG. These probes are useful tools to study the interaction of CTLRs with their ligands, with applications similar to those of antibodies, often in combination with widely available fluorescent antibodies targeting the Fc fragment (anti-hFc). In particular, Fc-Dectin-1 has been extensively used to study the accessibility of $\beta$-glucans at the surface of pathogenic fungi. However, there is no universal negative control for Fc-CTLRs, making the distinction of specific versus nonspecific binding difficult. We describe here 2 negative controls for Fc-CTLRs: a Fc-control constituting of only the Fc portion, and a Fc-Dectin-1 mutant predicted to be unable to bind $\beta$-glucans. Using these new probes, we found that while Fc-CTLRs exhibit virtually no nonspecific binding to *Candida albicans* yeasts, *Aspergillus fumigatus* resting spores strongly bind Fc-CTLRs in a nonspecific manner. Nevertheless, using the controls we describe here, we were able to demonstrate that *A. fumigatus* spores expose a low amount of $\beta$-glucan. Our data highlight the necessity of appropriate negative controls for experiments involving Fc-CTLRs probes.

**IMPORTANCE** While Fc-CTLRs probes are useful tools to study the interaction of CTLRs with ligands, their use is limited by the lack of appropriate negative controls in assays involving fungi and potentially other pathogens. We have developed and characterized 2 negative controls for Fc-CTLRs assays: Fc-control and a Fc-Dectin-1 mutant. In this manuscript, we characterize the use of these negative controls with zymosan, a $\beta$-glucan containing particle, and 2 human pathogenic fungi, *Candida albicans* yeasts and *Aspergillus fumigatus* conidia. We show that *A. fumigatus* conidia nonspecifically bind Fc-CTLRs probes, demonstrating the need for appropriate negative controls in such assays.

**KEYWORDS** C-type lectin receptors, pattern-recognition receptor (PRRs), pathogen-associated molecular patterns (PAMPs), host-pathogen interaction, *in vitro* assays

C-type lectin receptors (CTLRs) constitute a superfamily of transmembrane proteins, with some of them acting as Pattern Recognition Receptors (PRRs) that recognize Pathogen-Associated Molecular Patterns (PAMPs) (1). Although CTLRs can bind to PAMPs from bacteria, viruses, and parasites, a large body of work has focused to characterize their function in recognizing fungi and initiating antifungal immune responses. During an encounter with fungi, CTLRs bind to components of the fungal cell wall, a complex and dynamic structure at the interface between the fungal cell and its environment (2). The composition of the cell wall is highly variable among fungal species, yet some components are ubiquitously present in the wall of most human fungal pathogens, such as $\beta$-glucan, a polysaccharide

Address correspondence to Janet A. Willment, j.willment@exeter.ac.uk, or Gordon D. Brown, gordon.brown@exeter.ac.uk.

The authors declare no conflict of interest.

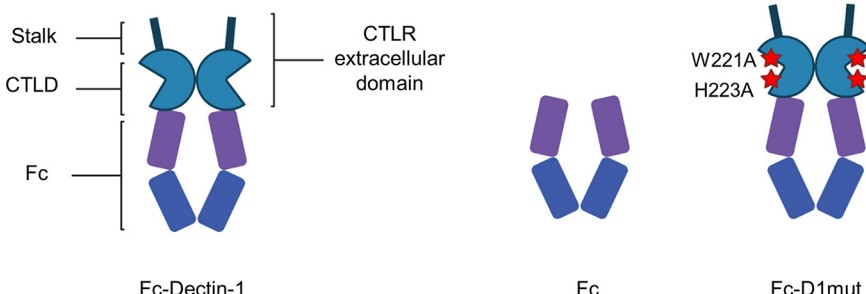

**FIG 1** Cartoon representation of Fc-proteins used in this study. Fc-CTLRs are soluble dimers constituted of the constant fraction (Fc) of the human IgG fused to the C-terminal end of the stalk and the C-type lectin-like Domain (CTLD), representing the full extracellular domain of CTLRs. As controls for Fc-Dectin-1 (left), we developed a Fc-control (center), constituted only of the Fc portion, and a Fc-D1mut (right), bearing W221A and H223A mutations and predicted unable to bind $\beta$-glucans.

recognized by Dectin-1 (encoded by *CLEC7A*) (3). Binding of $\beta$-glucans by transmembrane Dectin-1 induces the internalization of the target and its degradation into the phagolysosome, as well as the generation of Reactive Oxygen Species (ROS) and the secretion of multiple inflammatory mediators (4, 5). The essential role of Dectin-1 in initiating an antifungal immune response is illustrated by the increased susceptibility to fungal infections in individuals carrying polymorphisms of *CLEC7A* (6, 7). Polymorphisms in genes encoding for other CTLRs are also associated with increased susceptibility to invasive fungal infections (8). It is therefore essential to understand how the mammalian immune system recognizes fungi in order to predict and prevent susceptibility to fungal infections.

When studying the interaction between CTLRs and fungi, a key area of focus is determining the nature and accessibility of fungal PAMPs. Indeed, the exposure of CTLR ligands can be growth-stage specific (9). For instance, previous work showed that PAMPs from *Aspergillus fumigatus* conidia are largely protected from recognition by the so-called rodlet layer constituted of the hydrophobin RodA but become exposed when conidia start to swell and germinate, or when *rodA* is genetically deleted (10, 11). Commonly used tools for the aforementioned studies are Fc-CTLRs fusion proteins (12). Fc-CTLRs are constituted of the extracellular domain of CTLRs (responsible for ligand binding) fused to a mutated version of the human IgG constant fraction domain (Fc), which is unable to bind to Fc receptors or activate the complement (11, 13). Fc-CTLRs can be produced cheaply and in abundance by purification from transfected mammalian cell lines, and can be used like antibodies, in applications such as flow cytometry or immunofluorescence microscopy. However, the field lacks appropriate negative controls, often using an anti-human Fc secondary antibody only as a control (14–16), which does not factor in potential nonspecific background binding of Fc-CTLRs. Therefore, the aim of this study was to develop and characterize appropriate negative controls for these studies, using Dectin-1 as an exemple.

## RESULTS

**Construction of negative controls.** We devised 2 negative controls. The first control we constructed was a Fc-control that does not contain any CTLR extracellular domain (13, 17) (Fig. 1). We also constructed a mutated Fc-Dectin-1, Fc-D1mut, containing 2 point-mutations in the $\beta$-glucan-binding site of Dectin-1: W221A and H223A, which have been previously shown to abolish $\beta$-glucan binding (16). While the Fc-control allows determination of the nonspecific binding caused by the Fc fragment, Fc-D1mut provides a CTLR probe that lacks ligand binding activity.

**Negative controls constructs do not bind to $\beta$-glucan or intact *C. albicans* cells.** We first tested the ability of the purified Fc-control protein (the Fc-control region devoid of any CTLR extracellular domain) to recognize $\beta$-glucans by flow cytometry, using depleted zymosan, a $\beta$-glucan-rich cell wall preparation of *Saccharomyces cerevisiae* treated to eliminate Toll-Like Receptors binding properties (4). We also measured binding of the anti-hFc antibody alone, which has been previously used as a negative control for Fc-CTLR staining of fungi in other studies (14–16). The Fc-control did not bind zymosan, when compared to

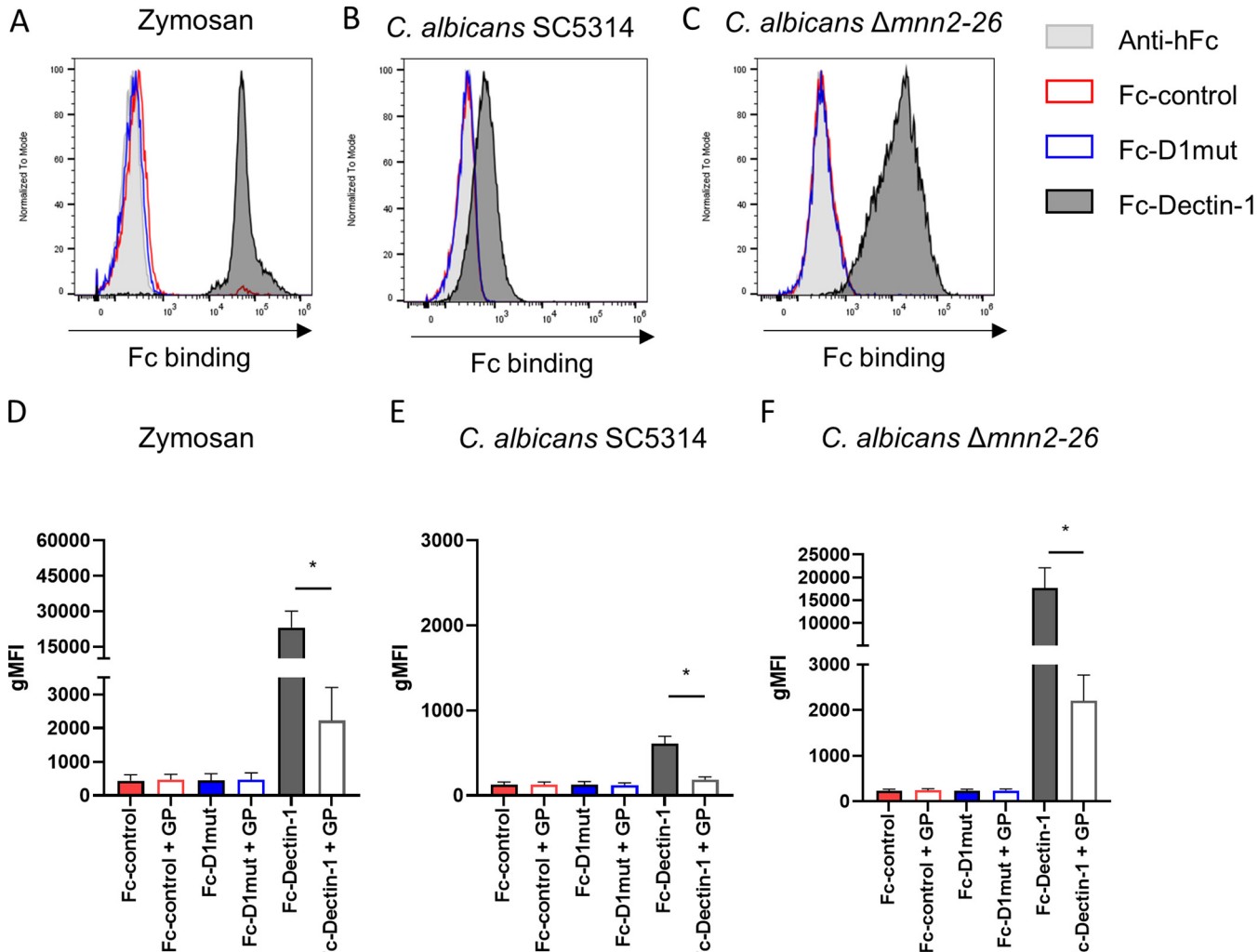

**FIG 2** Fc-control and Fc-D1mut do not bind zymosan or *C. albicans* yeasts. (A), (B), and (C) Representative histograms of Fc-CTLR staining of zymosan (A), *C. albicans* SC5314 (B), and *C. albicans* Δ*mnn2-26* (C). Zymosan, SC5314, and Δ*mnn2-26 C. albicans* yeasts were stained with Fc-control (red), Fc-D1mut (blue), Fc-Dectin-1 (black), and/or the secondary anti-Fc (gray) and the fluorescent intensity was measured by flow cytometry. (D), (E), and (F) Geometric mean fluorescent intensity (gMFI) resulting from Fc-CTLR binding to zymosan (D), SC5314 yeasts (E), or Δ*mnn2-26* yeasts (F) in presence or absence of 10 μg/mL glucan-phosphate (GP) inhibition. (D) to (F) Data pooled from 5 experiments represented as mean and SEM. Statistical differences were calculated by Mann-Whitney U-test, *, *P* value < 0.05.

the anti-hFc secondary, indicating that the Fc-control does not recognize these fungal particles (Fig. 2A and Fig. S1A). In contrast, Fc-Dectin-1 bound zymosan strongly, as expected (17) (Fig. 2A and Fig. S1A). We next tested the mutated Fc-Dectin-1 (Fc-D1mut), which was also unable to recognize zymosan (Fig. 2A and Fig. S1A). Thus, both Fc-control and Fc-D1mut are unable to bind β-glucan rich particles and could represent appropriate negative controls for β-glucan staining of fungal cells.

We then tested whether Fc-control and Fc-D1mut could be used as negative controls to stain live yeasts of *Candida albicans*. Fc-control and Fc-D1mut did not recognize *C. albicans* SC5314, with levels of binding similar to that of anti-hFc secondary antibody (Fig. 2B and Fig. S1B). In contrast, Fc-Dectin-1 bound to *C. albicans*, although at low levels (Fig. 2B and Fig. S1B), as reported previously (18). Much higher binding of Fc-Dectin-1 was observed with the Δ*mnn2-26* mutant of *C. albicans*, in which β-glucans are highly exposed due to the absence of *N*-mannan side chains in the cell wall (18, 19) (Fig. 2C, Fig. S1C). Fc-control and Fc-D1mut did not bind to the Δ*mnn2-26* mutant (Fig. 2C and Fig. S1C). Therefore, the Fc-control and Fc-D1mut represent good negative controls for Fc-Dectin-1 recognition of zymosan or *C. albicans* yeasts.

We also sought to confirm β-glucan staining specificity by Fc-Dectin-1 by a competition assay, in which Fc-proteins were incubated with glucan-phosphate (GP, a soluble β-glucan

able to bind Dectin-1 [20]) prior to the staining of fungi. GP inhibited Fc-Dectin-1 binding to zymosan, *C. albicans* SC531, and *C. albicans* Δ*mnn2-26* (Fig. 2D to F), demonstrating that Fc-Dectin-1 binding to the fungal particles is β-glucan specific. GP did not affect the staining of either *C. albicans* or zymosan by Fc-control or Fc-D1mut (Fig. 2D to F). Overall, our results show that Fc-control and Fc-D1mut are appropriate negative controls for Fc-Dectin-1 staining of β-glucans on zymosan or *C. albicans* yeasts.

**Appropriate controls allow detection of β-glucans in *A. fumigatus* spores using Fc-Dectin-1 despite high nonspecific binding.** We next tested whether Fc-control and Fc-D1mut could be used as negative controls for the staining of another human fungal pathogen and chose resting spores of *Aspergillus fumigatus* (21). It is well established that β-glucans are poorly exposed in resting spores due to the presence of the rodlet layer and dihydroxynaphthalene-melanin (DHN-melanin) (10, 11, 16). For these experiments, we used the *A. fumigatus* ku80 strain, a laboratory strain unable to perform non-homologous end joining but used as a reference for wild-type cell wall composition (16). Notably, we detected considerable nonspecific binding of both the Fc-control and Fc-D1mut proteins to *A. fumigatus* resting conidia, compared to the anti-hFc secondary alone (Fig. 3A and B). The binding of Fc-Dectin-1 to the spores was significantly higher than the binding of Fc-control and Fc-D1mut (Fig. 3B), suggesting potential β-glucan exposure in resting Spores. We confirmed that Fc-Dectin-1 binding of the spores was due to β-glucan recognition as we could inhibit the binding using GP (Fig. 3C). Of note, GP did not have statistically significant effect on Fc-control and Fc-D1mut binding to the conidia. Thus, using appropriate controls, our data suggest that low levels of β-glucans are exposed at the surface of resting conidia and accessible for recognition by Fc-Dectin-1.

We also tested our Fc constructs using conidia of *A. fumigatus* cell wall mutants known to expose more β-glucans than ku80: Δ*pksP* (lacking DHN-melanin), Δ*rodA* (lacking the rodlet layer), and Δ*rodA,pksP* (lacking both) (16). Notably, both Fc-control and Fc-D1mut demonstrated variable degrees of binding to the mutant conidia (Fig. S2A to F), showing that the relative binding of Fc-proteins (measured in our experiments as gMFI) cannot be used to compare β-glucan exposure in different strains without the appropriate controls. Similarly to what we observed in our other experiments, Fc-Dectin-1 binding could be inhibited by GP, validating recognition of exposed β-glucans by Fc-Dectin-1 (Fig. S2G to I), confirming previous observations (10, 16).

Because it is generally thought that resting *A. fumigatus* conidia mask exposed PAMPs (10), we sought to validate our observation of β-glucan exposure on resting wild type *A. fumigatus* conidia using a different approach. For this, we made use of BWZ.36 reporter cells that express a chimeric receptor consisting of the transmembrane and extracellular domains of a CTLR of interest, fused with the intracellular domain of the CD3ζ chain (22) (Fig. S3A). We validated that BWZ-Dectin-1 (expressing the CD3ζ/Dectin-1 chimeric receptor) were activated by β-glucans by treating the cells with zymosan or fixed *C. albicans* Δ*mnn2-26*, which have high levels of exposed β-glucan, as shown by our earlier experiments (Fig. S3B). By contrast, neither zymosan nor fixed *C. albicans* Δ*mnn2-26* yeasts activated the BWZ.36 parental cell line or BWZ/D1mut cells (expressing a CD3ζ/Dectin-1mutant chimeric receptor) (Fig. S3B). In order to determine whether β-glucans could be detected at the surface of ku80 conidia by the reporter cell system, we incubated BWZ.36 parental control, BWZ-D1mut and BWZ-Dectin-1 mutant cells with fixed ku80 resting conidia. In agreement with our previous results with Fc-Dectin-1, we observed that BWZ-Dectin-1, but not the control or BWZ-D1mut cells, were activated in a dose-dependent manner by ku80 resting conidia (Fig. 3D), confirming the presence of exposed β-gucans at the surface of *A. fumigatus* resting conidia.

## DISCUSSION

Fc-CTLRs constitute useful tools to study the interaction between CTLRs and their ligands. As such, Fc-Dectin-1 has been widely used as a tool to study β-glucan exposure at the cell wall surface of fungi by several methods, including flow cytometry and microscopy (14, 16, 18). However, there are often no appropriate negative controls for such experiments, in which a fluorescent secondary antibody is used as negative control (12, 14). Therefore, we

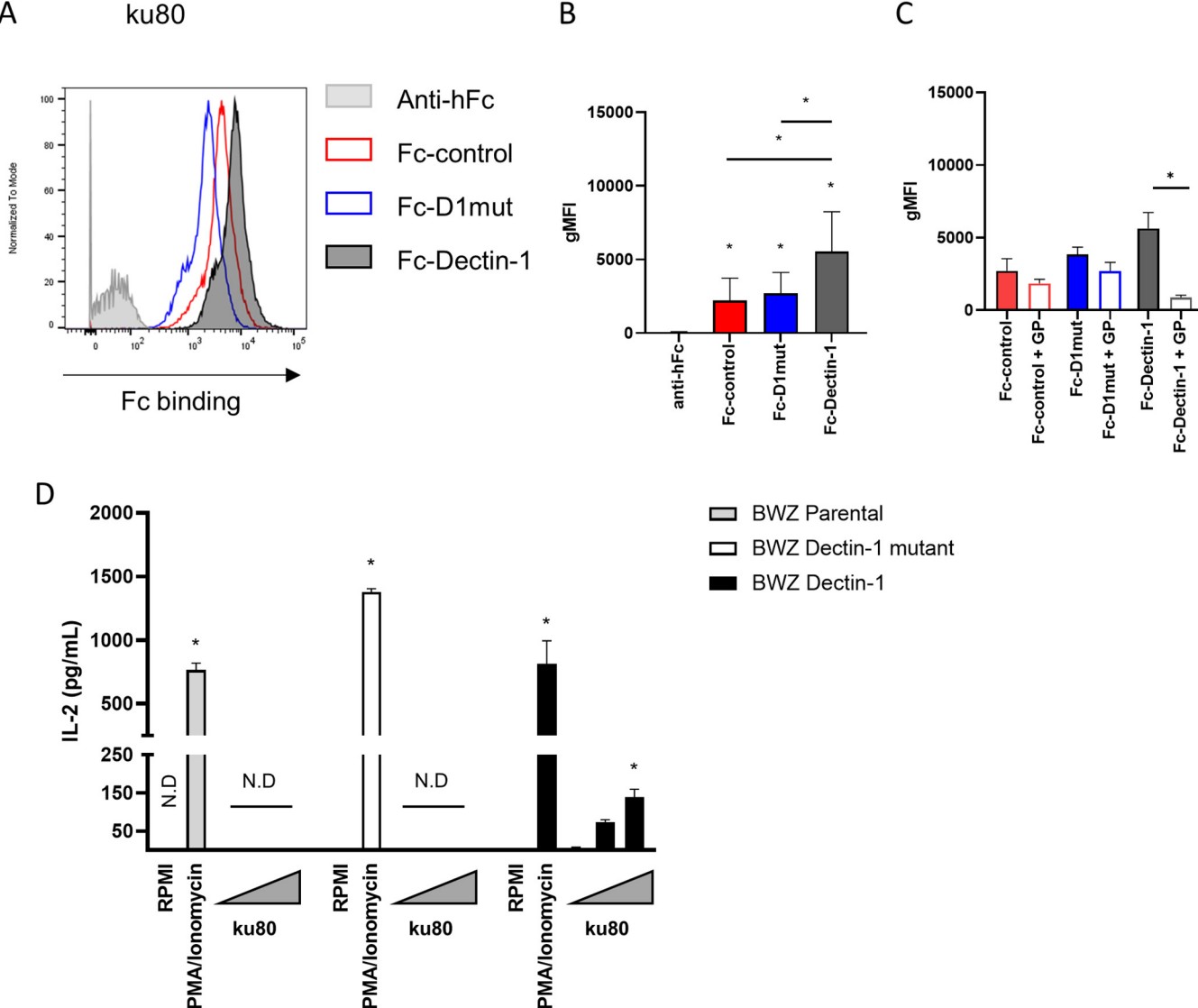

**FIG 3** *A. fumigatus* ku80 resting conidia expose β-glucans. (A) Representative histograms of Fc-CTLR staining of ku80 resting conidia labeled with Fc-control (red), Fc-D1mut (blue), Fc-Dectin-1 (black), and/or anti-Fc (gray). (B) gMFI of Fc-CTLR staining of ku80 conidia, 8 independent experiments represented as mean and SEM. (C) gMFI resulting from Fc-CTLR binding in presence or absence of glucan-phosphate (GP) inhibition. Data from 4 experiments represented as mean and SEM. (B) and (C) Statistical differences were calculated by Mann-Whitney U-test, *, $P$ value < 0.05. Statistical differences between anti-hFc and Fc-probe are indicated by the star above bar. (D) BWZ.36 reporter cells were incubated overnight with PMA/Ionomycin or formaldehyde-fixed ku80 resting conidia at increasing MOIs. IL-2 secretion was measured by ELISA. Pooled data are from 4 independent experiments represented as mean and SEM. Statistical differences were determined by Kruskal-Wallis. *, $P$ value < 0.05; N.D., not detected.

produced 2 controls for Fc-CTLR staining: a soluble Fc-control domain and a Fc-D1mut, in which the β-glucan binding site is mutated in order to prevent recognition of the ligand (23). We showed that neither the Fc-control nor the Fc-D1mut bound to β-glucan-rich zymosan particles or *C. albicans* yeasts, with a binding intensity similar to that of the anti-hFc secondary antibody only (Fig. 2). In contrast, we found that both the Fc-control and Fc-D1mut significantly bound *A. fumigatus* conidia in a nonspecific manner (Fig. 3 and Fig. S2). This shows that using secondary only anti-hFc is an inappropriate negative control for analyzing PAMPs exposure in some fungi. The cell wall of fungi is rich in proteins with lectin properties (i.e., able to bind sugars) (24) which could be responsible for the nonspecific binding of the Fc-CTLRs which are likely to be glycosylated. In particular, mammalian cells lines (such as HEK293T, in which our Fc-CTLRs are produced), are known to heavily glycosylate proteins (25). Differences in carbohydrate affinity by fungal lectins from *C. albicans* and *A. fumigatus* could thus underlie the difference of nonspecific binding by these fungi.

Although it is widely accepted that *A. fumigatus* conidia are poorly immunogenic due to the limited PAMPs exposure (21), our results using Fc-Dectin-1 (along with the appropriate negative controls) clearly show that resting conidia can be recognized by Dectin-1, a PRR essential for antifungal immune responses (7, 26). We validated this observation using reporter cell lines. While a low amount of $\beta$-glucan exposed at the conidial surface has been described previously (27), our observations contrast with several other studies (11, 16, 28, 29). Of note, some of these studies used soluble Dectin-1 (conjugated with the Fc fragment or not) without a soluble CTLRs control (11, 16, 29). Nevertheless, some could confirm the absence of exposed $\beta$-glucans using cellular assays (29). Variations in growth conditions are one possibility underlying these differences. It is unlikely that this is a strain-specific result, as we found $\beta$-glucans exposed on the resting conidia of several *A. fumigatus* reference strains such as ATCC13073, CEA10, and Af293 (data not shown).

Overall, our results show the importance of appropriate negative controls when studying the interaction of fungi with CTLRs. Some fungi, such *A. fumigatus* conidia shown here, can bind nonspecifically to Fc- fusion proteins. Although we did not observe such background binding to wild type SC5314 and Δmnn2-26 *C. albicans* yeasts, this should be considered when using other *C. albicans* morphotypes, cell wall mutants, or other *Candida* spp. Fc-CTLRs remain a powerful tool to study the interaction between pathogens and PRRs, but we strongly recommend inclusion of appropriate negative controls in assays involving this type of fusion proteins. Additionally, alternative assays such as inhibition of binding by purified ligands to confirm specificity and/or the use of reporter cells to demonstrate binding in a cellular context could constitute complementary approaches that would reinforce conclusions regarding the interaction between fungi and PRRs.

## MATERIALS AND METHODS

**Fungal culture and preparation.** *Aspergillus fumigatus* ku80, Δ*pksP*, Δ*rodA*, and Δ*rodA,pksP* (16) cultures were prepared from frozen glycerol stocks and grown on Potato Dextrose Agar (Difco) for 7 days at 37°C in the dark. Resting conidia were harvested in water and filtered through a 40 $\mu$m cell strainer to remove mycelium fragments. Conidia were washed 3 times with water, resuspended in PBS, aliquoted and frozen at −20°C until staining. *Candida albicans* SC5314 and Δ*mnn2-26* (18) were cultivated from frozen glycerol stock onto YPD plates. A single colony was cultured overnight in liquid YPD at 30°C with shaking at 150 rpm. Yeasts were centrifuged, washed 3 times with water and resuspended in PBS before staining. For reporter cell stimulation, fungi were fixed with 1% formaldehyde overnight at 4°C. The next day, fixed fungi were washed three times with water and resuspended in PBS.

**Construction of pSecTag Fc-fusion plasmids.** pSecTag Fc and pSecTag Fc-Dectin-1 were constructed previously (17, 30). To mutate Dectin-1, Dectin-1 extracellular domain was amplified from pSecTag Fc-Dectin-1 using 5′-AAAGGTACCTAGCATTTTGGCGACA-3′ and 5′-AAAGAATTCCAGTTCCTTCTCACAGAT-3′. The mutations were introduced by 2 different PCRs amplifying the 5′ (5- AAAGGTACCTAGCATTTTGGCGACA-3′ and 5′- CTCTGATCCAGCAATCGCTACACAATT-3′, with mutated bases underlined) or the 3′ (5′-AATTGTGTA GCGATTGCTGGATCAGAG-3′ and 5′-AAAGAATTCCAGTTCCTTCTCACAGAT-3′) sequence of Dectin-1 extracellular domain. Both fragments were then fused by PCR using the first pair of primers and cloned in pSecTag Fc by digestion/ligation.

**Preparation of Fc-proteins.** Fc-control, Fc-Dectin-1, and Fc-D1mut were prepared as described before (17). In brief, HEK293T were transfected with pSecTag containing the construct for each Fc-control protein, selected with Zeocin (Invivogen), and cultured in DMEM GlutaMAX-I (Gibco) supplemented with 5% Fetal Calf Serum (FCS), 1% Penicillin/Streptomycin, and 25 mM HEPES at 37°C 5% $CO_2$. Fc-proteins were purified from culture supernatants by Protein A-Sepharose (GE) and dialyzed against PBS. As the predicted molecular weight of the Fc-control protein is considerably smaller, concentrations of Fc-control and Fc-D1mut were normalized to the concentration of Fc-Dectin-1. For this, 1 $\mu$g/mL Fc-Dectin-1 and serial dilutions of Fc-control or Fc-D1mut in carbonate buffer were coated onto Maxisorp Immunoplate overnight at 4°C. The next day, the plate was washed, and horseradish peroxidase-conjugated anti-human (Jackson ImmunoResearch) was added at a 1:10,000 dilution and incubated 45 min at room temperature. The plate was washed and 100 $\mu$L of TMB substrate (Pierce) was added, and the reaction was stopped with 100 $\mu$L 2 M $H_2SO_4$. Concentrations of each Fc-control proteins required to achieve equivalent absorbance of Fc-Dectin-1 at 5 $\mu$g/mL were then calculated and used for each Fc-CTLRs in staining experiments.

**Fc-protein staining.** A total of 1 $\mu$g depleted zymosan (Invivogen), $10^6$ *A. fumigatus* conidia, or *C. albicans* yeasts were resuspended in 50 $\mu$L Flow Buffer (PBS [no Ca²⁺, no Mg²⁺, Gibco], 1.5% Bovine Serum Albumin fraction V [BSA, Roche], 5 mM EDTA [Invitrogen]), in a 96-well V-bottom plate and incubated on ice for 30 min. A total of 50 $\mu$L Fc-Dectin-1, Fc-control, or Fc-D1mut were directly added to a final concentration of 5 $\mu$g/mL or equivalent concentration. For inhibition assays, Fc-control proteins were mixed with 40 $\mu$g/mL glucan-6-phosphate (final concentration 20 $\mu$g/mL) for 30 min in PBS at room temperature and centrifuged 5 min 13,000 × *g* to pellet potential precipitates prior to addition to the conidia. Fc-proteins were incubated with fungal particles or zymosan for 40 min on ice, then particles were washed three times with Flow Buffer,

resuspended in 100 $\mu$L Flow Buffer 1% formaldehyde, and incubated overnight at 4℃. The next day, conidia were washed three times, and resuspended in 100 $\mu$L Flow buffer containing R-PE-conjugated donkey anti-human IgG (Jackson ImmunoResearch) at a final dilution of 1:200. Conidia were incubated 30 min in the dark at room temperature, washed three times, resuspended in 150 $\mu$L and acquired on a BD Accuri flow cytometer. Data were analyzed on FlowJo v10.6.2.

**Construction of pMXs CD3$\zeta$/Dectin-1 mutant.** pMXs and pMXs-CD3$\zeta$/Dectin-1, a plasmid encoding for the chimeric protein constituted of the CD3$\zeta$ intracellular chain and Dectin-1 extracellular domain were described previously (22). The 3' sequence of Dectin-1 mutant, carrying W221A and H223A, was amplified from pSecTag Fc-Dectin-1 mutant using 5'- TTACTGCACAATTGTGTAGC-3' and 5'- AATTGCGGCCGCTTACAGTT CCTTCTCACAGATAC-3', digested and ligated into pMXs-CD3$\zeta$/Dectin-1 in place of the 3' sequence of Dectin-1.

**Culture and stimulation of BWZ reporter cells.** BWZ.36 parental, BWZ-Dectin-1 (described earlier [22]), and BWZ-Dectin-1 mutant were cultured in RPMI GlutaMAX-I (Gibco) supplemented with 10% Fetal Calf Serum (FCS), 1% Penicillin/Streptomycin, and 25 mM HEPES at 37℃ 5% $CO_2$. Cells were maintained under hygromycin (400 $\mu$g/mL), and BWZ Dectin-1 and BWZ D1mut were additionally maintained under puromycin (4 $\mu$g/mL selection). For stimulation, cells were lifted with TryPLE (Gibco), washed, and counted. A total of $4\times10^4$ cells were seeded in a 96-well TC-treated flat-bottom plate and incubated with culture medium or culture medium supplemented with 40 ng/mL phorbol 12-myristate 13-acetate (PMA, Sigma) and 1.5 $\mu$g/mL ionomycin (Sigma), 50 $\mu$g/mL depleted zymosan (Invitrogen), or fixed fungi at a multiplicity of infection (MOI) of 5, 10, or 20 overnight at 37℃ 5% $CO_2$. Supernatant was harvested and the concentration of IL-2 was determined by ELISA as per manufacturer's protocol (BD OptEIA mouse IL-2).

**Statistics.** All statistics analysis were conducted in GraphPad Prism 9.0. For comparison of gMFI in fungal particle staining, normal distribution was tested by Shapiro-Wilk test. Statistical differences between groups were calculated by Mann-Whitney U-test in order to mimic an analysis with a single control each time. For IL-2 secretion by BWZ cells, normal distribution was tested by Shapiro-Wilk. Statistical differences between test groups and RPMI (unstimulated) conditions by Kruskal-Wallis test. In all graphs, $P$ values $<$ 0.05 are represented by an asterisk.

## SUPPLEMENTAL MATERIAL

Supplemental material is available online only.
**SUPPLEMENTAL FILE 1**, TIF file, 0.7 MB.
**SUPPLEMENTAL FILE 2**, TIF file, 1.2 MB.
**SUPPLEMENTAL FILE 3**, TIF file, 0.9 MB.

## ACKNOWLEDGMENTS

We acknowledge funding from the Wellcome Trust (102705, 217163), the MRC Centre for Medical Mycology at the University of Exeter (MR/N006364/1; MR/N006364/2) and the NIHR Exeter Biomedical Research Centre. The views expressed are those of the author(s) and not necessarily those of the NIHR or the Department of Health and Social Care.

This work was also supported, in part, by RO1GM119197 from the National Institutes of Health. For the purpose of open access, the author has applied a CC BY public copyright license to any Author Accepted Manuscript version arising from this submission.

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
