## [Reviewer comments · Microbiology Spectrum]

Microbiology Spectrum

Development of negative controls for Fc-C-type lectin receptor probes

Remi Hatinguais, Madalaine Kay, Fabian Salazar Lizama, Daniel Conn, Peter Cook, David Williams, Janet Willment, and Gordon Brown

Corresponding Author(s): Gordon Brown, University of Exeter

Review Timeline:

Submission Date:	March 15, 2023
Editorial Decision:	April 2, 2023
Revision Received:	April 11, 2023
Accepted:	April 14, 2023

Editor: James Konopka

Reviewer(s): The reviewers have opted to remain anonymous.

Transaction Report:

DOI: <https://doi.org/10.1128/spectrum.01135-23>

April 2, 2023

Dr. Gordon D Brown
University of Exeter
Exeter
United Kingdom

Re: Spectrum01135-23 (Development of negative controls for Fc-C-type lectin receptor probes)

Dear Gordon:

The reviewers agree that this manuscript will have a significant impact on the field. The mutant forms of the Fc-Dectin-1 that are unable to bind to beta-glucans will improve rigor by their use as negative controls. The reviewers have only minor comments.

Thank you for submitting your manuscript to Microbiology Spectrum. As you will see your paper is very close to acceptance. Please modify the manuscript along the lines I have recommended. As these revisions are quite minor, I expect that you should be able to turn in the revised paper in less than 30 days, if not sooner. If your manuscript was reviewed, you will find the reviewers' comments below.

When submitting the revised version of your paper, please provide (1) point-by-point responses to the issues raised by the reviewers as file type "Response to Reviewers," not in your cover letter, and (2) a PDF file that indicates the changes from the original submission (by highlighting or underlining the changes) as file type "Marked Up Manuscript - For Review Only". Please use this link to submit your revised manuscript. Detailed instructions on submitting your revised paper are below.

Link Not Available

Sincerely,

James Konopka

Reviewer comments:

Reviewer #1 (Comments for the Author):

This is an interesting and an important study wherein the authors demonstrated the importance of negative controls in immunoassays pertaining to pattern recognition receptors. I have only two minor questions, one is general and one related to the technique.

1. This study performed with two major fungal pathogens, *Candida albicans* and *Aspergillus fumigatus*. With *C. albicans*, there was no non-specific binding, while *A. fumigatus* shows high non-specific binding. Can the authors discuss a bit more on this subject?
2. In the Material and Methods section, it is described that after fixing with 1% PFA overnight, the fungal samples were washed with PBS before taking them for stimulating study. As the authors may be aware, PFA crosslinks two proteins and some PFA may be having only one binding partner having other arm of the PFA with free reactive group, which may have an impact on immunostimulation. Therefore, after PFA fixation, introducing a quenching step will be important. Did the authors try quenching PFA fixed fungal samples before taking them for BWZ-reporter cell assay?

Reviewer #2 (Comments for the Author):

The G. Brown lab's generation of Fc-Dectin-1 chimeric proteins many years ago was a seminal step in the ability to detect exposed beta-glucans using immuno-detection methods. However, because the Fc-Dectin-1 molecules are not true antibodies, non-immune serum could not be used as a negative control to show binding specificity. To remedy this, Hatinguias et al. constructed mutant forms of the Fc-Dectin-1 that are unable to bind to beta-glucans (either lacking the binding domain or mutating key residues required for beta-glucan binding). The results are verified with multiple fungal particles providing scientific rigor. Intriguingly, the authors demonstrate that the negative control Fc-Dectin-1 molecules can bind to *Aspergillus* conidia in a beta-glucan independent manner suggesting some inherent "stickiness" of conidia. This however is unexplored. The results will be useful for the fungal field.

Suggestions

The D1mut construct is particularly useful as a negative control for the Fc-Dectin-1 probe, but should not be extrapolated to use as the negative control for other Fc-CTLR constructs (e.g., those binding mannan). The authors should acknowledge this in the discussion

The authors should at least speculate on the nature of the binding of the D1mut to *Aspergillus* resting conidia. Since the Fc-only construct showed a similar level of binding to conidia, the binding of Fc-D1mut appears to be independent of the lectin region. Do antibodies also display a low level of non-specific binding to conidia? Would the inclusion of low levels of detergent reduce dispersion force interactions sufficiently to reduce it?

line 76 add "carbohydrate-recognizing" extracellular domain of CTRLs to provide better context for those unfamiliar with the C-type lectin receptor structure

line 147 beta-glucan exposure is relatively low and should be qualified as such

line 190 typo Fig. 12 should be Fig. 2

line 267, 302 define "depleted zymosan"

Preparing Revision Guidelines

Please return the manuscript within 60 days; if you cannot complete the modification within this time period, please contact me. If you do not wish to modify the manuscript and prefer to submit it to another journal, please notify me of your decision immediately so that the manuscript may be formally withdrawn from consideration by Microbiology Spectrum.

Dear reviewers,

Thank for taking the time to review our manuscript. You will find in this letter a point-by-point response to your comments.

Reviewer #1

1. This study performed with two major fungal pathogens, Candida albicans and Aspergillus fumigatus. With C. albicans, there was no non-specific binding, while A. fumigatus shows high non-specific binding. Can the authors discuss a bit more on this subject?

It is a good point that we should have discussed indeed, especially as both reviewers have asked about it. We have now commented on this observation in the Discussion section of the manuscript.

2. In the Material and Methods section, it is described that after fixing with 1% PFA overnight, the fungal samples were washed with PBS before taking them for stimulating study. As the authors may be aware, PFA crosslinks two proteins and some PFA may be having only one binding partner having other arm of the PFA with free reactive group, which may have an impact on immunostimulation. Therefore, after PFA fixation, introducing a quenching step will be important. Did the authors try quenching PFA fixed fungal samples before taking them for BWZ-reporter cell assay?

Yes, fixation quenching (with glycine for instance) would have be a good step to include in our protocol. We did not quench the PFA in either the staining protocol or the stimulation, which might be a concern although the fixative is extensively washed. Prior to the stimulation, samples are prepared in batch in RPMI 10% FCS medium that contain a number of proteins and amino acids, including glycine, any of them might be react with potential free arms of the PFA, and therefore would to prevent the fixation of the PFA to any receptor at the surface of the cells. These proteins/amino acid could also affect the stimulation of the cells. However, we have also observed activation of the BWZ Dectin-1 by heat-inactivated conidia and also $\Delta pyrG$ conidia (background A1160) that are unable to germinate in the absence of uracil and uridine, without the need to inactivate the spores, which is always likely to affect the outcome

of such assay. Therefore, we do not think that omitting the quenching of the fixation is likely to have a strong biological effect in this context.

Reviewer #2

The D1mut construct is particularly useful as a negative control for the Fc-Dectin-1 probe, but should not be extrapolated to use as the negative control for other Fc-CTRL constructs (e.g., those binding mannan). The authors should acknowledge this in the discussion

We have now made this explicit. Although we believe that using either the Fc-control or the Fc-D1mut as negative control is better than having none, we agree that ideally specific negative controls, with point-mutation(s) in the carbohydrate-binding site would constitute ideal controls for Fc-CTRL assays.

The authors should at least speculate on the nature of the binding of the D1mut to Aspergillus resting conidia. Since the Fc-only construct showed a similar level of binding to conidia, the binding of Fc-D1mut appears to be independent of the lectin region. Do antibodies also display a low level of non-specific binding to conidia? Would the inclusion of low levels of detergent reduce dispersion force interactions sufficiently to reduce it?

We have now added a paragraph in the discussion to address the difference in non-specific binding of Fc-CTRLs by *C. albicans* and *A. fumigatus*. We believe this difference lies in potential differences in the production of fungal lectins by the two fungi, which could lead to different carbohydrate-binding specificity (Fc-CTRLs are glycosylated). We have not checked whether antibodies generally bind to the spores, although from these experiments we only see a marginal non-specific binding of the fluorescent anti-human Fc (as estimated by the low gMFI compared with unstained conidia). We have not tried to add detergent at step of the staining protocol.

line 76 add "carbohydrate-recognizing" extracellular domain of CTRLs to provide better context for those unfamiliar with the C-type lectin receptor structure

We added "ligand binding" to take into account non-carbohydrate ligands.

line 147 beta-glucan exposure is relatively low and should be qualified as such

Modified.

line 190 typo Fig. 12 should be Fig. 2

Corrected, thanks for the careful reading.

line 267, 302 define "depleted zymosan"

The term is now defined line 101 when zymosan is mentioned first in the results section.

We thank again the reviewers for their time and hope that we addressed their comments appropriately.

The authors.

April 14, 2023

Dr. Gordon D Brown
University of Exeter
Exeter
United Kingdom

Re: Spectrum01135-23R1 (Development of negative controls for Fc-C-type lectin receptor probes)

Dear Gordon:

Your manuscript has been accepted, and I am forwarding it to the ASM Journals Department for publication. You will be notified when your proofs are ready to be viewed.

Sincerely,

James Konopka
Editor, Microbiology Spectrum
